# Drop Jump Performance Improves One Year Following Anterior Cruciate Ligament Reconstruction in Sportsmen Irrespectively of Psychological Patient Reported Outcomes

**DOI:** 10.3390/ijerph20065080

**Published:** 2023-03-14

**Authors:** Claudio Legnani, Matteo Del Re, Giuseppe M. Peretti, Vittorio Macchi, Enrico Borgo, Alberto Ventura

**Affiliations:** 1Sport Traumatology and Minimally Invasive Surgery Center, IRCCS Istituto Ortopedico Galeazzi, 20161 Milan, Italy; 2IRCCS Istituto Ortopedico Galeazzi, 20161 Milan, Italy; 3Department of Biomedical Sciences for Health, University of Milan, 20133 Milan, Italy

**Keywords:** knee, anterior cruciate ligament, ACL reconstruction, outcomes, psychological readiness, vertical jump, drop jump

## Abstract

Our study aims to prospectively report the functional outcomes of 31 sportsmen following anterior cruciate ligament (ACL) reconstruction, up to 12 months after surgery, with regards to subjective tests and drop jump performance, and to investigate the correlations between these variables, to be used for determining the return to sports after ACL reconstruction. Lysholm score, Tegner activity level, and the ACL–Return to Sport after Injury (ACL-RSI) scale were evaluated preoperatively, at 6 months, and at 12 months after surgery. Drop vertical jump was recorded using an infrared optical acquisition system. Lysholm and ACL-RSI scores significantly improved at the 12-month follow-up compared to the baseline and 6-month evaluations (*p* < 0.001). Concerning Tegner activity level, no statistically significant differences were reported between pre- and post-operative status (*p* = 0.179). Drop jump limb symmetry index significantly improved at 12 months, with the mean value improving from 76.6% (SD: 32,4) pre-operatively to 90.2% (SD: 14.7; *p* < 0.001) at follow-up. Scarce positive correlation was reported between the ability to perform drop jumps and activity level in athletes one year after ACL reconstruction. In addition, subjective knee score and psychological readiness were not related to jumping performance.

## 1. Introduction

The rate of athletes sustaining anterior cruciate ligament (ACL) injuries while practicing sporting activity is growing; as a result, so is the number of ACL reconstructions performed worldwide [1,2]. Return to sporting activity is a major concern for patients undergoing knee surgery, although after ACL reconstruction only two-thirds of patients are able to resume sport at the same level they practiced before injury [3], and the rate of re-injury is relatively high, especially among younger athletes [4]. Reasons for that include the persistence of limb asymmetries related to knee explosive strength, proprioception, and psychological readiness, which altogether play a crucial role in allowing athletes to successfully return to a safe practice of sport [5,6]. To overcome these issues, subjective questionnaires with point scales, as well as functional tests, have been developed to propose reliable criteria for clearing athletes to return to sport after ACL reconstruction [7,8,9]. Commonly used test batteries evaluating lower limb function include hop tests (typically single-leg hop, triple hop, and side hop) and vertical jump tests [7]. Vertical jump tests are widely used to assess functional ability by measuring explosive strength, power, and reactivity [10,11], and demonstrated superior reliability in detecting functional deficits related to pathology compared to horizontal hop tests [10,11]. More recently, the role of the drop jump as a reliable tool to assess deficits in proprioception and balance following ACL reconstruction has been investigated [12]. It has been reported that explosive strength and propulsion force differences between limbs persist among athletes resuming sporting activities after ACL surgery [12]. Deficits in drop jump landing ability have also been linked to fear of reinjury and kinesiophobia [13]. It has also been suggested that limb kinetic asymmetries recorded during drop vertical jumps may be due to compensatory strategies used to reduce the load on the affected limb [14]. Therefore, it seems particularly significant that drop jump performance should be carefully assessed in order to allow health practitioners to clear athletes for the return to sport after surgery.

The purpose of this study was to prospectively report the functional outcomes of sportsmen following ACL reconstruction up to 12 months after surgery, with regards to subjective tests and drop jump performance, and to investigate the correlations between these variables, to be used for determining the return to sports after ACL reconstruction.

## 2. Materials and Methods

### 2.1. Patient Recruitment

Thirty-five consecutive patients suffering from ACL rupture who underwent ACL reconstruction between January and December 2021 were included in this prospective observational study. Only patients practicing sports who had undergone primary ACL reconstruction, aged between 18 and 45 years at the time of surgery, were included in the present research. Only male patients were recruited in order to avoid potential gender-related bias regarding muscle mass. Subjects were excluded in the case that they had past history of knee joint surgery; concomitant treatment of chondral pathology, or multi-ligamental surgery. Variables such as age, gender, body mass index (BMI), and time from injury to surgery were recorded (Table 1). 

The study was approved by the Institutional Review Board of the IRCCS San Raffaele Hospital, Milan, Italy (IRB number: 57/INT/2020). Informed consent was obtained from all participants prior to data collection.

### 2.2. Surgical Technique and Rehabilitation Protocol

Arthroscopic-assisted ACL reconstruction using autologous hamstring graft was performed in all patients as previously described [4]. All patients underwent the same rehabilitation protocol, including immediate regaining of knee extension and isometric exercises; walking with crutches with partial weight bearing was allowed for the first 3 weeks. Swimming and indoor cycling were started 3 months after surgery. Jump technique training and plyometric exercises were introduced 5 months after operation.

### 2.3. Follow-Up Assessment

Patients were asked to complete three subjective questionnaires: Lysholm, Tegner, and ACL-return to sports after injury (ACL-RSI) preoperatively, at 6 months, and at 12 months after surgery. Drop vertical jump test was performed at the same time points as described by Noyes et al. [15]. The patient dropped from a box, at a height of 30 cm, and immediately performed a vertical jump as high as possible after hitting the ground, keeping the hands on the hips. Each jump was repeated three times. During each trial, the test was performed with the uninjured limb first, followed by the injured. The mean value of the three tests was included for analysis. Data were recorded using the OptoGait system (Microgate, Bolzano, Italy), an infrared optical acquisition device which allows the detection of jump height from flight time. The Limb Symmetry Index (LSI) calculated represented the percentage of test performance on the uninvolved limb compared to the healthy limb. Patients’ evaluations were performed using the same equipment.

### 2.4. Statistical Analysis

Statistical analyses were performed using Graphpad Prism v8.0 (Prism Software, La Jolla, CA, USA). Differences were tested in accordance with the result of the Shapiro-Wilk test for data distribution. The Mann–Whitney-U-test was used in case of non-normal data distribution, while a two-way ANOVA with repeated measures was used for normally-distributed data. Spearman’s correlation coefficient was calculated to determine correlations between jump LSI and scores. Recorded *p* values < 0.05 were considered statistically significant. Post hoc power analysis was conducted, evaluating improvements in Lysholm score after surgery. For a test with alpha = 0.05, the study sample size provided a test power > 90%.

## 3. Results

Four patients (11%) were lost at follow-up. Twelve months after surgery, 31 patients were evaluated. The mean time from injury to surgery was 2.8 months (SD: 1.1). Patients’ demographics and anthropometric data are reported in detail in Table 1. 

### 3.1. Subjective Knee Function

Functional scores significantly improved at 12 months follow-up compared to the baseline and 6-month evaluations (*p* < 0.001). Improvements were also observed at 6 months with respect to the baseline for Lysholm and ACL-RSI scores. Concerning Tegner activity level, no statistically significant differences were reported between the pre-operative and 6-month evaluations (*p* = 0.179) (Table 2).

### 3.2. Drop Jump Test

Drop jump for the healthy knee showed improvements at the 12-month evaluation compared to the 6-month evaluation (*p* < 0.01), but not between the 6-month follow-up and the baseline (*p* = n.s.). Drop jump for the injured knee showed improvements at the 12-month evaluation compared to the baseline (*p* < 0.001) and the 6-month follow-up (*p* < 0.001). No differences were observed between the 6-month evaluation and the baseline. Drop jump LSI showed improvements at the 12-month evaluation compared to the baseline (*p* < 0.001), with the mean value improving from 76.6% (SD: 32,4) to 90.2% (SD: 14.7; *p* < 0.001) at follow-up. No differences were observed between the 6-month evaluation and the baseline, or between the 6- and 12-month evaluations (*p* = n.s.) (Figure 1).

### 3.3. Correlations between Drop Jump LSI and Subjective Point Scales

At 12 months, correlation between drop jump LSI and subjective scores demonstrated weak positive correlation with Tegner activity level (*p* < 0.05). Weak correlation with ACL-RSI was also found, but it was not statistically significant (*p* = 0.08). No correlations with the Lysholm score were observed (*p* = 0.25) (Table 3).

## 4. Discussion

Functional tests and patient-reported outcome measures (PROMs) represent useful tools to apply in clinical practice in order to assess surgical outcomes and patients’ ability to return to sport and recreational activities. 

According to our findings, functional scores significantly improved at 12 months follow-up after ACL reconstruction compared to the baseline and 6-month evaluations. Improvements were also observed at 6 months with respect to the baseline for Lysholm and ACL-RSI scores. Concerning Tegner activity level, no statistically significant differences were reported between pre- and post-operative status.

Correlation between jumping ability and subjective perception has been previously analyzed [15,16,17,18]. According to our findings, 12 months after surgery, weak positive correlation was reported between drop jump LSI and Tegner activity level. Interestingly, 12 months after surgery, no correlation between jumping performance and either Lysholm or ACL-RSI scores was observed. 

According to Roe et al., 6 months after ACL reconstruction, the performance of a single leg step-down test was strongly correlated to PROMs [19]. Lee et al. found significant correlations between single-leg vertical jump test, ACL-RSI, Tegner, and IKDC scores [20]. In a previous paper using the same cohort of patients 6 months following ACL surgery, patients with higher ACL-RSI scores reported better functional and clinical outcomes and improved vertical jump performance at 6 months following ACL surgery [18]. Conversely, Tavares et al. investigated health-related quality of life following ACL reconstruction and found few to no correlations between this variable and performance in hop tests [21]. Reporting on 303 patients up to 12 months after ACL reconstruction, Gauthier et al. failed to find any correlation between psychological readiness to return to sport and functional ability [22]. 

Although some authors found a significant correlation between PROMs and functional ability, our study results reflect the lack of interaction between these variables, as previously reported in literature. This absence of evidence may outline the limit of self-reported questionnaires in the analysis of subjects’ physical capacity.

Bakal et al. compared inter-limb asymmetry while performing drop vertical jump between adolescents who had undergone ACL reconstruction and a healthy group. They found that asymmetry was significantly more pronounced in the ACL reconstruction group [14].

According to our findings, average drop jump LSI at 6 months after surgery was 76.6%, while this value reached the threshold of 90% one year after ACL reconstruction. LSI > 90% is usually suggested as the cutoff score to allow patients to resume sporting activity at a recreational level following ACL surgery [23], although its reliability is questioned, since risks of performance overestimation exist due to concomitant worsening of function in the uninvolved limb [24]. Many papers have recommended that clearance for return to sports should be nine months or later from surgery [25,26,27]. 

Read et al. reported on 370 professional sportsmen and detected between-limb asymmetries in eccentric and concentric loading parameters after 9 months following ACL reconstruction, thus supporting the occurrence of a compensatory mechanism to offload the involved limb during the vertical jump [28]. 

Our results may confirm that the return to sport should be shifted to at least 6 months after surgery, which is usually considered the cutoff value for allowing sport resumption.

Our study possesses limitations. A trend for correlation between drop jump LSI and ACL-RSI was found, but failed to demonstrate statistical significance. A larger sample size could have had more power and thus be more sensitive to demonstrating a small difference between variables. In addition, jumping capacity is affected by many parameters which were not the focus of the present research, and we acknowledge that the use of the drop jump to investigate neuromuscular restoration following ACL reconstruction should be added as a limitation. Additionally, the use of PROMs and the lack of a control group of healthy individuals constitute further limitations. Long-term prospective follow-up studies with larger cohorts are required to corroborate these findings.

## 5. Conclusions

Drop jump LSI significantly improved at 12 months after surgery, while scarce positive correlation was reported between the ability to perform drop jump and activity level in athletes one year after ACL reconstruction. In addition, subjective knee scores and psychological readiness were not related to jumping performance.

## Figures and Tables

**Figure 1 ijerph-20-05080-f001:**
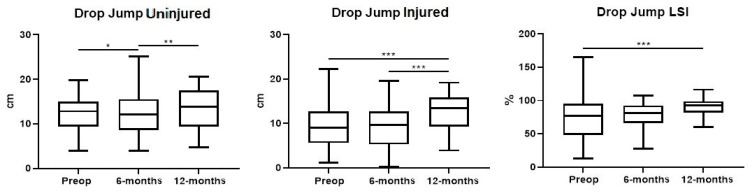
Box-plots showing differences in outcome scores from pre-operative evaluation to 6-month follow-up after surgery. The black line inside the box represents the median value. The lowest bar represents the minimum value, the bottom and top of the boxes represent the interquartile range (25th and 75th percentiles), and the top bar represents the maximum value. Points outside the limits represent outliers * *p* = n.s.; ** *p* < 0.05; *** *p* < 0.001.

**Table 1 ijerph-20-05080-t001:** Patient demographics and anthropometric data.

No. of Patients	31
Male gender	31
Mean age at surgery (SD) (yr)	34.3 (11.1)
Mean BMI (SD)	25.8 (3.0)

SD: standard deviation; BMI: Body Mass.

**Table 2 ijerph-20-05080-t002:** Overall comparison between pre-operative and follow-up status.

	Pre-Operative	6 Months	12 Months	*p*-Value
Mean Lysholm score (SD)	69.7 (14.7)	87.2 (11.2)	94.0 (11.7)	<0.001
Mean ACL-RSI score (SD)	45.1 (23.3)	69.5 (15.5)	82.6 (20.7)	<0.001
Median Tegner activity level (range)	4 (2–9)	5 (2–9)	6 (5–9)	<0.001

*p*-value refers to difference between baseline and follow-up. SD: standard deviation; ACL-RSI: Anterior Cruciate Ligament Return to Sport after Injury.

**Table 3 ijerph-20-05080-t003:** Correlation between drop jump LSI and point scales.

	DJ LSI vs. Lysholm	DJ LSI vs. ACL-RSI Score	DJ LSI vs. Tegner
Spearman r	0.2053	0.3076	0.3474
95% confidence interval	−0.1518 to 0.5192	−0.04724 to 0.5978	0.02591 to 0.6435
*p* value (two-tailed)	0.2564	0.0815	0.0327

DJ: Drop Jump; LSI: Limb symmetry index; ACL-RSI: Anterior Cruciate Ligament Return to Sport after Injury.

## Data Availability

Raw data will be provided on request.

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
