# Peer review of "Drop Jump Performance Improves One Year Following Anterior Cruciate Ligament Reconstruction in Sportsmen Irrespectively of Psychological Patient Reported Outcomes"

_ijerph, 2023, doi:10.3390/ijerph20065080_

Round 1

Reviewer 1 Report

Let me start by congratulating you on your excellent work. In a very important theme and that it is quite interesting to find solutions to this problem. However, I think that the introduction presents few investigations on the subject and leaves the reader a bit in the void of information.

Half of the discussion is the presentation of the results. Only the final part is to discuss what was found in the results and in other studies by other authors. The description of the methodology could also be better described, such as, for example, what is the height of the step? Did all patients have the same form of operation, but were they all in the same rehabilitation program? With or in the same doctor/physiatrist office?

Finally, I believe that the inclusion of two girls may interfere a little and be different, not only but also because of the muscle mass that may be involved in recovery. Hence, I judged that it would be more favorable not to include these two girls.

Reviewer 2 Report

First of all, thank you very much for allowing me to review this very interesting manuscript.

Material and methods

- I suggest you justify why you have used the flight time to calculate the jump height, instead of the take-off speed as the most reliable method.

Results

-You should center the gender categories in Table 1.

-They should justify why the female sample is so small. According to the literature, women have a higher risk of this injury so it seems odd that there are many more men.

-I suggest you include the p-values in table 2.

-Also, the indications of which section is the mean and which is the standard deviation can be difficult to understand. You could include it in a top row.

-Please review the formatting of all table footers.

Discussion

-At the beginning of the discussion, the term PROMs appears, which has not been previously defined.

-
